# Leptin as a Key Player in Insulin Resistance of Liver Cirrhosis? A Cross-Sectional Study in Liver Transplant Candidates

**DOI:** 10.3390/jcm9020560

**Published:** 2020-02-19

**Authors:** Iva Košuta, Anna Mrzljak, Branko Kolarić, Marijana Vučić Lovrenčić

**Affiliations:** 1Department of Gastroenterology, Merkur University Hospital, Zajčeva 19, 10000 Zagreb, Croatia; anna.mrzljak@gmail.com; 2School of Medicine, University of Zagreb, Šalata 3, 10000 Zagreb, Croatia; 3Department of Epidemiology, Andrija Štampar Teaching Institute of Public Health, Mirogojska cesta 16, 10000 Zagreb, Croatia; branko.kolaric@stampar.hr; 4Faculty of Medicine, University of Rijeka, Ul. Braće Branchetta 20/1, 51000 Rijeka, Croatia; 5Department of Clinical Chemistry and Laboratory Medicine, Merkur University Hospital, Zajčeva 19, 10000 Zagreb, Croatia; vucic@idb.hr

**Keywords:** liver cirrhosis, insulin resistance, leptin, adiponectin

## Abstract

Insulin resistance is associated with increased risk of death and liver transplantation in the cirrhotic population, independent of disease aetiology. However, factors accounting for insulin resistance in the context of cirrhosis are incompletely understood. This study aimed to investigate the association between adiponectin and leptin with insulin resistance in cirrhotic patients and to assess the influence of disease severity on insulin resistance and metabolic status. This cross-sectional study included 126 non-diabetic cirrhotic transplant candidates. The homeostasis model assessment 2 model was used to determine the insulin resistance index, and fasting adiponectin, leptin, insulin, c-peptide, glucose, HbA1c, and lipid profiles were analysed. Insulin resistance was detected in 83% of subjects and associated with increased leptin, fasting plasma glucose and body mass index, and lower triglyceride levels. Logistic regression analysis identified leptin and triglycerides as independent predictors of insulin resistance (OR 1.247, 95% CI 1.076–1.447, *p* = 0.003; OR 0.357, 95% CI 0.137–0.917, *p* = 0.032.). Leptin levels remained unchanged, whereas adiponectin levels increased (*p* < 0.001) with disease progression, and inversely correlated with HbA1c (ρ = −0.349, *p* < 0.001). Our results indicate that leptin resistance, as indicated by elevated leptin levels, can be regarded as a contributing factor to insulin resistance in cirrhotic patients, whereas triglycerides elicited a weak protective effect. Progressively increasing adiponectin levels elicited a positive effect on glucose homeostasis, but not insulin sensitivity across disease stages.

## 1. Introduction

When higher circulating insulin levels are necessary to achieve glucose-lowering responses, a subject is considered insulin resistant [1]. To overcome insulin resistance and maintain normal metabolic functions, insulin secretion is increased, leading to a state of compensatory hyperinsulinemia [2]. This metabolic dysfunction is associated with a cluster of abnormalities with serious clinical consequences, including type 2 diabetes and cardiovascular disease [3,4]. Furthermore, IR has been implicated in the advancement of chronic liver disease, via enhancement of liver steatosis and fibrosis [5]. In chronic hepatitis C virus (HCV) infection, IR was identified as a risk factor for advanced/severe steatosis and associated with significant fibrosis independently from steatosis [6,7,8].

An association between liver cirrhosis, the final common pathway of chronic liver disease, and insulin resistance was described by Megyesi et al., who also coined the phrase ‘hepatogenous diabetes’, emphasizing the role of the liver in disturbances of glycemic control [9]. IR tends to be an early event in the course of the disease, possibly explained by insufficient clearance of insulin due to reduced hepatocellular function. Both disease progression with further hepatocellular dysfunction and portal hypertension-related portosystemic shunting of insulin aggravate hyperinsulinemia [10,11]. Insulin resistance then occurs via insulin receptor desensitization and downregulation due to chronic hyperinsulinemia [11,12,13]. Transition to overt diabetes mellitus is primarily driven by beta-cell dysfunction and the inability to compensate for the insulin resistance [14].

Independently of disease aetiology, IR in cirrhosis appears to identify a subgroup of patients with a worse prognosis. IR might have a role in the initial steps contributing to the development of clinically significant portal hypertension, most likely due to alterations in the regulation of hepatic endothelial nitric oxide [15]. Furthermore, IR was associated with the development and recurrence of hepatocellular carcinoma (HCC) in patients with chronic HCV infection, regardless of the presence of diabetes [16,17]. Nkontchou et al. showed that IR was independently associated with an increased risk of death or liver transplantation in a cohort of 249 chronic HCV patients with compensated cirrhosis [18]. However, factors accounting for IR in the context of cirrhosis are still largely unknown. Studies in the general population implicate adipocytokines, metabolically active products of adipose tissue, in energy homeostasis and insulin sensitivity/resistance. The best-studied adipocytokines include adiponectin and leptin. Adiponectin levels are negatively correlated with insulin resistance and diabetes mellitus, suggesting a causative role in the development of insulin resistance [19]. Leptin, a modulator of insulin secretion and sensitivity, is implicated in the inflammatory and fibrogenic pathways in the liver [20,21,22,23]. Still, in the cirrhotic patient, data pertaining to the role of adipocytokines in the development of insulin resistance is sparse, with most studies focusing on adipocytokine levels compared to the healthy population [24]. Accordingly, increased systemic adiponectin was demonstrated in cirrhosis, independent of disease etiology, age, or body mass index (BMI) [25,26]. Data regarding leptin concentrations is far more inconsistent, with increased, reduced or unchanged leptin levels reported [27,28,29,30]. Regarding the effect of the adipocytokines on IR, Kaser et al. found no correlation between adiponectin levels and insulin resistance in 87 patients with liver cirrhosis of different aetiologies [31]. Nkontchou et al. found that in patients with compensated HCV cirrhosis, insulin resistance but not serum levels of adiponectin and leptin predicted the occurrence of HCC and of liver-related death or transplantation, but no attempt of correlation between IR and adipocytokine levels was made [18].

This study aimed to investigate the association between adiponectin and leptin with insulin resistance in patients with liver cirrhosis, as well as to assess the influence of disease severity on insulin resistance and metabolic status.

## 2. Patients and Methods

### 2.1. Patients and Study Design

This cross-sectional study was carried out at the Liver Transplant Center, Merkur University Hospital (Zagreb, Croatia). One hundred and twenty-six adult cirrhotic patients being evaluated for liver transplantation from December 2013 to April 2018 were included. Cirrhosis was diagnosed according to histological or clinical criteria, including the presence of complications such as variceal haemorrhage, ascites and/or hepatic encephalopathy and/or imaging and laboratory exams consistent with the diagnosis. Liver disease severity was determined by the Child–Pugh scoring system and model for end-stage liver disease (MELD), applicable to all disease etiologies [32,33]. The Child–Pugh score is determined by scoring five clinical measures of liver disease. A score of 1, 2, or 3 is given to each measure, with three being the most severe. The five clinical measures include total bilirubin, serum albumin, INR, ascites, and hepatic encephalopathy. Patients scoring ≥7 Child–Pugh points, i.e., classified as Child–Pugh class B or C, were considered to have a decompensated disease stage.

Inclusion criteria were compensated and decompensated cirrhosis regardless of disease aetiology. Exclusion criteria were prior solid-organ transplantation due to the possible confounding effect of immunosuppressant drugs, and previously established diagnosis of diabetes with or without anti-diabetic therapy and/or fasting plasma glucose (FPG) ≥7.0 mmol/L at the time of evaluation.

The study was performed according to the principles of the declaration of Helsinki and was approved by the Merkur University Hospital’s Ethics Committee on December 30, 2013. Written informed consent was obtained from each patient.

### 2.2. Anthropometric Measurements and Laboratory Analysis

All subjects were subjected to extensive history taking (including presenting complaint, history of presenting complaint, past medical history, drug history, known allergies, family history, and social history) and complete clinical evaluation. Baseline characteristics collected included age, gender, height, weight, and BMI.

Weight was measured using a balance beam scale (Detecto 339 Balance Beam Scale, Webb City, MO, USA) with light clothing without shoes and expressed in kilograms, and height was measured using a wall-mounted stadiometer and expressed in centimeters in order to calculate BMI.

Fasting venous blood samples were collected for determination of lipid profile (total cholesterol, HDL-cholesterol, LDL-cholesterol, and triglycerides (TGC)), liver biochemistry including aspartate aminotransferase (AST), alanine aminotransferase (ALT), gamma-glutamyl transferase (GGT), alkaline phosphatase (ALP), liver function (total bilirubin and international normalised ratio (INR)). After clotting, the sera were separated by centrifugation and stored at −70 °C until the analysis of adipocytokines (conducted in November 2016 from samples collected in December 2013, and in August 2018 for samples collected from November 2013 until the conclusion of the study). Lipid profile, glucose, AST, ALT, GGT, ALP, and bilirubin were measured using routine laboratory methods on an automated analytical platform (AU680, Beckman Coulter, Brea, CA, USA). Plasma INR was derived from prothrombin time results determined by an automated coagulation analyzer (Sysmex 2100i, Siemens Healthineers, Marburg, Germany). HbA1c was measured in EDTA-anticoagulated whole blood samples using an automated turbidimetric inhibition immunoassay (HbA1c Gen 3, Cobas Integra 400 Plus, Roche Diagnostic, Basle, Switzerland), traceable to the International Federation of Clinical Chemistry and Laboratory Medicine (IFCC) reference system and reported in National Glycohemoglobin Standardization Program (NGSP) units (%). Fasting insulin and c-peptide were determined by automated chemiluminescent immunoassays (Advia Centaur XP, Siemens Healthineers, Tarrytown, NY, USA). Adiponectin and leptin concentrations were determined by validated enzyme-immunoassay methods (Biovendor, Czech Republic).

### 2.3. Estimate of Insulin Resistance

The homeostasis model assessment HOMA2 calculator (version 2.2.2, Diabetes Trials Unit, University of Oxford, available at http://www.dtu.ox.ac.uk/homacalculator/index.php) was used to estimate the insulin resistance index (IRI) from fasting glucose and insulin concentrations [34]. The HOMA model is a mathematical method for estimation of beta-cell function (HOMA2-%B) and insulin resistance (HOMA2-IRI) from basal (fasting) plasma glucose and insulin concentrations. An updated HOMA2 calculator, used in this study, provides reliable estimates of model parameters for fasting plasma glucose concentration up to 20.0 mmol/L and specific insulin concentration ranging from 20–300 pmol/L. The HOMA2 model is a non-linear model accounting for variations in hepatic and peripheral glucose resistance, as well as circulating proinsulin levels. A value for insulin sensitivity expressed as HOMA2-%S can also be obtained from this model, but this parameter is simply the reciprocal of HOMA2-IR.

### 2.4. Data Analysis and Statistics

The normality of distribution was assessed by the Shapiro–Wilk test. Non-parametric tests were used for variables with non-Gaussian distribution. Spearman’s rank correlation coefficient was used to assess possible associations between different parameters. Logistic regression analysis was used to determine independent risk factors for IR in cirrhosis. IR was treated as a binary variable, with insulin resistance index (IRI) values >1.7 considered as insulin resistance. The cut-off used was established as the upper 75th percentile of HOMA2-IR in a validated study with a matching Central-European population [35]. Considering the lack of harmonization in IRI cut-off due to biological and analytical variability, an evidence-based cut-off on a continuum of insulin resistance index variable, considered as optimal determinant for metabolic syndrome, was applied [36]. Adjustments were performed for age and gender. The level of statistical significance was chosen to be 0.05. Statistical analysis was performed by Stata/IC ver. 14.2, StataCorp LLC and MedCalc Statistical Software version 18.11.6 (MedCalc Software bvba, Ostend, Belgium; https://www.medcalc.org; 2019).

## 3. Results

### 3.1. Study Population

A total of 126 patients were included in the study. Mean age at the time of inclusion was 57.6 ± 8.9 years, and 72% were male. The most frequent aetiology of cirrhosis was alcoholic liver disease (56%), followed by viral hepatitis (22%). Decompensated cirrhosis was present in 89%, and HCC in 28% of subjects. Baseline clinical and biochemical characteristics of the patients are given in Table 1 and Table 2.

### 3.2. Adiponectin and Leptin

The mean adiponectin concentration of the study cohort was 16.25 mg/L. Significant differences were observed between liver disease stages (Figure 1A), while there was no difference observed between the insulin-resistant and non-insulin-resistant sub-group (Table 2). Adiponectin levels correlated with the MELD score (ρ = 0.327, *p* < 0.001) and were inversely correlated with HbA1c (ρ = −0.349, *p* < 0.001). The mean leptin concentration of the study cohort was 7.55 ug/L. Significant differences were observed between the insulin-resistant and non-insulin-resistant subjects (Table 2), whereas there was no difference between Child–Pugh categories (Figure 1B). Leptin levels correlated with insulin resistance index (IRI) (ρ = 0.484, *p* < 0.001), BMI (ρ = 0.347, *p* < 0.001) and insulin concentrations (ρ = 0.336, *p* < 0.001).

### 3.3. Insulin Resistance

Median IRI was 2.5 ± 2.3, and 83% of subjects were insulin resistant according to the applied cut-off (>1.7). As shown in Table 2, the insulin-resistant group had significantly higher levels of leptin, BMI and FPG and lower levels of triglycerides compared to the non-insulin resistant group. A weak but significant correlation was found between IRI and BMI (ρ = 0.300, *p* = 0.001), FPG (ρ = 0.354, *p* < 0.001), and HbA1c (ρ = 0.202, *p* = 0.023).

Logistic regression analysis identified leptin (OR 1.25, *p* = 0.003) and triglyceride (OR 0.36, *p* = 0.032) concentrations as independent risk factors for IR in cirrhosis, while age and gender were found to be biologic confounders, as shown in Table 3. The logistic regression model was significant; chi^2^ = 57.49, *p* < 0.01. The model explained 57% of the variance in IR and correctly classified 90% of cases.

### 3.4. Liver Disease Stage

No significant differences in leptin, FPG, insulin, and IRI were observed across the Child–Pugh stages. Adiponectin and c-peptide were significantly higher (*p* < 0.01 and *p* = 0.01, respectively), while triglycerides, cholesterol, and HbA1c were lower (*p* < 0.01 for all three variables) with an increase in Child–Pugh stage (Figure 1, Table 4). Furthermore, a moderately strong correlation between MELD score and c-peptide concentrations was found in both insulin-resistant patients and the whole population (ρ = 0.573, *p* < 0.005 and ρ = 0.559, *p* < 0.001, respectively).

## 4. Discussion

In this study, we aimed to: (i) investigate the association between adiponectin and leptin with insulin resistance in patients with liver cirrhosis, and (ii) assess the influence of disease severity on insulin resistance and metabolic status. Our results reveal a high prevalence of insulin resistance in patients with liver cirrhosis clinically evaluated for liver transplantation. More than 80% of study subjects were found to be insulin resistant, with the median IRI being 3.3. The disease stage did not influence the intensity of insulin resistance as it was evenly distributed between the Child–Pugh stages. The insulin-resistant group of cirrhotic patients had significantly higher levels of leptin, insulin, BMI and FPG, and lower levels of triglycerides compared to the non-insulin resistant group. However, only leptin was identified as a risk factor for the occurrence of insulin resistance in our study population, indicating the possible association of hyperleptinemia with insulin resistance independently of age, gender, and body constitution. Triglyceride levels were established as a protective factor for insulin resistance in our patients.

Previous evidence implicates liver cirrhosis as an insulin-resistant state with insulin resistance as an early event in the natural history preceding the onset of cirrhosis [9,18]. Our results confirm these findings, but, to our knowledge, this is the first study to identify such a cluster of metabolic factors related to insulin resistance in patients with liver cirrhosis awaiting transplantation. The design of our study does not allow the presumption of a causal relationship(s). However, the association between hyperleptinemia and insulin resistance, and lack of protective effect of adiponectin, despite its increasing levels with progression of liver disease, provide some valuable insight into the metabolic aspects of cirrhosis in this specific clinical population.

Several mechanisms of hyperinsulinemia in chronic liver disease have been postulated. The most pronounced is an insufficient insulin clearance due to reduced hepatocellular function, with healthy liver parenchyma being replaced by fibrotic tissue [10,11]. Liver clearance of insulin is further reduced with the advancement of disease (i.e., the transition from compensated to decompensated liver cirrhosis as designated by Child–Pugh classification), due to portal-hypertension-related portosystemic shunting [37]. In addition, nonalcoholic fatty liver disease is often complicated with increased adiposity, which leads to hyperinsulinemia both in the non-cirrhotic and cirrhotic population [38]. Chronic hyperinsulinemia then leads to insulin resistance via the desensitization and downregulation of insulin receptors [11,12,13]. As it was already mentioned, insulin resistance is both an early process in the natural history of liver disease, often preceding the onset of liver cirrhosis, and a remarkable risk factor of death or transplantation [9,18].

A well-known mediator of insulin synthesis and action is the adipocytokine leptin, a 16 kDa protein hormone synthesized mainly in adipose tissue, with significant regulatory roles in metabolic processes and energy homeostasis [39,40]. Circulating leptin levels are directly related to the amount of body fat, thereby reflecting the status of long-term energy stores. Furthermore, leptin levels fluctuate according to changes in calorie intake with a marked decrease during starvation [41,42]. Leptin is structurally related to pro-inflammatory cytokines and is considered as an established contributing factor to the fibrogenic process due to its mitogenic actions on activated stellate cells. Thus, leptin has attracted attention as a possible link between obesity and/or chronic inflammation and liver fibrosis [43]. However, the possible role of leptin concerning metabolic disarrangements, including insulin resistance in liver cirrhosis, is less known.

Leptin is considered a counterregulatory hormone of insulin action via several mechanisms: directly suppressing insulin secretion from pancreatic beta-cells, counter-regulating the anabolic actions of insulin, regulating the autonomic nervous system and peripheral tissue sensitivity to insulin in response to changes of body mass, and finally, acting as a gut peptide signaling satiety [44]. The adipo-insular axis, a bidirectional feedback loop between the pancreas and adipocytes is responsible for hindering adipogenic effects of insulin via counter-regulation by leptin [45]. Dysregulation of the adipo-insular axis in obesity leads to a rise of basal plasma levels of insulin and leptin and development of compensatory peripheral tissue resistance to both hormones. Although insulin and leptin resistance progressively increase with the advancement of adiposity, there is a temporal discontinuity between them, with hyperinsulinemia preceding and causing hyperleptinemia [45,46].

In our group of cirrhotic liver transplant candidates, leptin levels correlated with IR, BMI, and insulin concentrations, indicating several possible mechanisms of hyperleptinemia in the context of cirrhosis. First, it is possible that the elevation of leptin levels reflects an effect of adiposity-driven dysregulation of the adipo-insular axis regardless of liver disease. Indeed, our results indicate that insulin-resistant patients with cirrhosis had a higher BMI and leptin levels in comparison to insulin-sensitive ones. However, this straightforward assumption disregards well-known limitations of BMI as an index of body constitution in cirrhotic patients. Namely, one of the critical attributes of decompensated cirrhosis is abnormal portal circulation resulting in excessive body water accumulation in the form of ascitic fluid or anasarca [47]. Therefore, it is not possible to reliably determine whether increased BMI is due to obesity or retained water. Recently published clinical guidelines suggest that in the case of fluid retention, calculation of BMI should be performed using the patient’s dry weight [48]. We were not able to estimate dry-weight in this study. However, as decompensated cirrhosis with ascites was present in almost 90% of our patients, it would be inappropriate to regard increased BMI as a simple reflection of an elevated amount of adipose tissue. Although our insulin-resistant subjects had higher BMI than insulin-sensitive controls, the median BMI of 26.9 kg/m^2^ indicates that the patients were moderately overweight, rather than obese, defined as BMI values above 30 kg/m^2^ [48]. Furthermore, considering that leptin is secreted not only by adipose tissue but also by the gastric mucosa in response to food intake and insulin, hyperleptinemia could be regarded as a result of an increased production of gastric leptin elicited by hyperinsulinemia [49]. Regardless of the circulating leptin origin, which is not possible to distinguish by the methods used in this clinical study, our results clearly show that insulin resistance is independently associated with hyperleptinemia, indicating that liver cirrhosis is accompanied with leptin resistance as well. It should be emphasised that the conjunction between hyperleptinemia and insulin resistance was not dependent on the etiology of liver cirrhosis, which was alcohol-related in the majority of our patients. While insulin resistance plays an essential role in liver steatosis, fibrosis, and finally cirrhosis originating from NAFLD and hepatitis C infection, molecular mechanisms involved in the pathogenesis of alcohol-related cirrhosis do not involve insulin resistance [50]. We postulate, therefore, that hyperleptinemia in cirrhosis is a consequence of hyperinsulinemia, but that this is driven, at least in part by hepatocellular dysfunction and abnormal portosystemic shunting of insulin, and not (just) by the natural history of the illness or an increase in adiposity.

At the same time, we found that adiponectin, the most abundant adipokine with a well-known ameliorating effect on insulin resistance, had no protective effect on insulin resistance in cirrhotic patients, despite rising levels associated with the severity of disease. However, a negative correlation with HbA1c indicates an at least partly preserved effect of adiponectin on glucose homeostasis. Since we found no association with indices of insulin sensitivity and beta-cell function, this effect is most likely attributable to the specific effect of adiponectin on the hepatic glucose production via identified regulatory pathways for glucose and lipid metabolism at transcriptional levels [51,52].

On the other hand, serum triglycerides significantly decreased across Child–Pugh categories and were identified as a weak, but significant, protective factor against insulin resistance. This finding may seem confusing considering the well-known role of liver steatosis in the natural history of liver fibrosis and cirrhosis, as well as insulin resistance. However, our results are in accord with previous evidence on the decreased diacyl- and triacylglycerol levels in alcoholic cirrhosis, which was regarded as a result of decreased lipogenesis due to reduced synthetic capacity of the liver with the progression of cirrhosis [53]. In addition, a decrease in serum triglyceride levels in our patients across Child–Pugh stages may also be related to the progressive hyperadiponectinemia, according to previous reports implicating a role of adiponectin in fat loss and hypermetabolism in liver cirrhosis [24]. The protective effect of triglycerides and relatively higher triglyceride levels in insulin-sensitive patients (Table 2) may simply reflect a favorable role of maintained lipogenic function, which may mitigate insulin resistance by removing free fatty acids, recognized metabolic contributors to insulin resistance from the circulation [54].

Our study found no correlation between leptin, adiponectin, and lipid markers. Independently of adipocytokines, pancreatic beta-cell function was significantly intensified with the aggravation of cirrhosis, as evidenced by both elevation of c-peptide levels across Child–Pugh stages and positive correlation with MELD scores. This finding indicates a maintained compensatory mechanism towards insulin resistance in our patients, which may, at least partly, be responsible for the control of fasting glycaemia and insulin resistance with the progression of disease.

Our study has several limitations that need to be pointed out. As it was performed in a liver transplantation referral center, most patients included were at an advanced (i.e., decompensated) disease stage. Furthermore, most patients were male, and the leading etiology was alcoholic liver disease, all of which may have led to a selection bias. Insulin sensitivity/resistance was not assessed by euglycemic clamp, which is a gold standard in insulin sensitivity determination. The study lacks a control group of healthy and/or chronic liver disease patients, and the cross-sectional study design does not address the causal relationships. In addition, a lack of an exact measure of body constitution for cirrhotic patients can be regarded as an important study limitation.

Despite these limitations, our study offers some new insights into a complex and yet unclear role of various metabolic pathways associated with insulin resistance in liver cirrhosis. Our results indicate that elevated leptin, a marker of leptin resistance, can be regarded as an important contributing factor of insulin resistance in cirrhotic patients. A weak protective effect of triglycerides against insulin resistance was identified. Progressively increasing adiponectin levels elicited a positive impact on glucose homeostasis, but not insulin sensitivity across liver cirrhosis stages. Further research is needed to determine causal relationships between the observed metabolic factors and clinical outcomes of patients with liver cirrhosis.

## Figures and Tables

**Figure 1 jcm-09-00560-f001:**
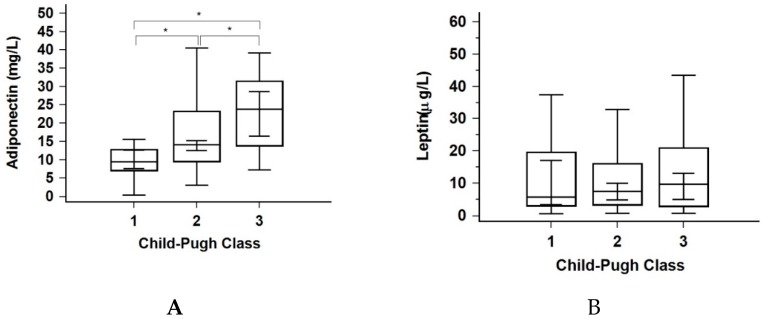
(**A**,**B**) Distributions of adiponectin and leptin through Child–Pugh classes. * *p* < 0.05.

**Table 1 jcm-09-00560-t001:** Number and percent of participants and incidence of manifestations (CP class = Child–Pugh class, SD = standard deviation).

Variable	N	%
Gender (Male:Female)	91:35:00	72:28:00
Etiology		
• Alcohol	69	54.8
• Viral	28	22.2
• Autoimmune	12	9.5
• Metabolic	1	0.8
• Cryptogenic	12	9.5
• Other	4	3.2
CP class		
• A	14	11.1
• B	43	34.1
• C	69	54.8
Insulin resistance	104	82.5
	Mean	SD
Age	57	9

**Table 2 jcm-09-00560-t002:** Characteristics of the patients with liver cirrhosis overall and according to insulin resistance (IRI cut-off 1.7).

Variable	Total (*n* = 126)		Insulin Resistant (*n* = 104)	Insulin Sensitive (*n* = 22)	
	Median ± IQR	Range	Median ± IQR	Median ± IQR	*p* *
Age (years)	58 ± 12	32–77	58 ± 12	56 ± 13	0.203
MELD score	15.43 ± 7	7–38	14.95 ± 7	16.44 ± 8	0.386
Leptin (ug/L)	7.55 ± 15.34	0.66–58.10	9.58 ± 16.18	3.00 ± 2.45	<0.001
Adiponectin (mg/L)	16.26 ± 18.04	3.01–137.17	16.95 ± 18.08	14.15 ± 15.58	0.951
BMI (kg/m^2^)	26.06 ± 5.81	15.94–44.19	26.86 ± 5.28	22.86 ± 5.01	<0.001
FPG (mmol/L)	5.3 ± 1.0	3.3–6.9	5.4 ± 0.9	4.63 ± 1.0	<0.001
HbA1c (%)	4.80 ± 0.6	3.4–6.0	4.8 ± 0.8	4.7 ± 0.5	0.163
Insulin (pmol/L)	113.0 ± 118.1	18.0–731.5	155.0 ± 115.6	68.1 ± 20.0	<0.001
C-peptide (nmol/L)	0.97 ± 0.66	0.39–5.32	1.05 ± 0.67	0.57 ± 0.44	<0.001
IRI	2.5 ± 2.3	0.40–15.90	3.3 ± 2.3	1.4 ± 0.45	<0.001
Triglycerides (mmol/L)	0.75 ± 0.59	0.28–3.88	0.72 ± 0.53	1.00 ± 1.14	0.035
Cholesterol (mmol/L)	3.2 ± 2.0	0.3–12.7	3.1 ± 2.0	3.35 ± 2.03	0.787
HDL-C (mmol/L)	0.91 ± 0.6	0.16–2.38	0.94 ± 0.63	0.8 ± 0.6	0.106
LDL-C (mmol/L)	2.00 ± 1.6	0.20–10.8	1.9 ± 1.7	2.1 ± 1.8	0.626
AST (IU/L)	63 ± 47	22–903	64 ± 41	56 ± 89	0.619
ALT (IU/L)	35 ± 36	10–241	36 ± 31	27 ± 69	0.458
AP (IU/L)	53 ± 65	13–1088	117 ± 78	108 ± 111	0.437
GGT (IU/L)	117 ± 82	42–567	54 ± 80	53 ± 72	0.529
Bilirubin (µmol/L)	53 ± 65	5–647	49 ± 63	74 ± 131	0.188
PV-INR	1.5 ± 1.3	1.0–3.0	1.5 ± 1.2	1.5 ± 1.3	0.672

* Mann–Whitney U test, IQR = interquartile range, MELD = model of end-stage liver disease, BMI = body mass index, FPG = fasting plasma glucose, IRI: insulin resistance index, HDL-C = high density lipoprotein cholesterol, LDL-C = low density lipoprotein cholesterol.

**Table 3 jcm-09-00560-t003:** Analysis of risk factors for insulin resistance in liver cirrhosis.

Variable	Odds Ratio	95% Conf. Interval	*p* *
CP score	0.954	0.777–1.171	0.651
Age	1.018	0.956–1.084	0.581
Gender	2.284	0.647–8.069	0.199
Leptin	1.247	1.076–1.447	0.003
Adiponectin	1.001	0.969–1.041	0.802
HbA1c	1.779	0.690–4.893	0.233
Triglycerides	0.357	0.139–0.917	0.032
Cholesterol	1.079	0.832–1.399	0.569
HDL-C	2.757	0.874–8.701	0.084
LDL-C	0.979	0.708–1.355	0.900

* Multiple logistic regression, CP = Child–Pugh, HDL-C = high density lipoprotein cholesterol, LDL-C = low density lipoprotein cholesterol.

**Table 4 jcm-09-00560-t004:** Metabolic characteristics of the patients according to stage of liver disease.

Variable	CP class A (*n* = 14)	CP class B (*n* = 43)	CP class C (*n* = 69)	*p* *
Age (years)	60.18 ± 11	56.56 ± 12	58.38 ± 12	0.321
MELD score	9.06 ± 2	13.77 ± 4	18.64 ± 8	<0.001
BMI (kg/m^2^)	26.13 ± 6.43	25.93 ± 7.26	26.24 ± 5.66	0.653
FPG (mmol/L)	5.3 ± 0.8	5.1 ± 1.2	5.3 ± 1.0	0.686
HbA1c (%)	5.2 ± 0.5	4.9 ± 0.7	4.6 ± 0.4	<0.001
Insulin (pmol/L)	127.85 ± 163.1	114.6 ± 117.9	110.8 ± 108.1	0.963
C-peptide (nmol/L)	0.78 ± 0.45	0.98 ± 0.77	1.03 ± 0.64	0.01
IRI	2.75 ± 3.35	2.5 ± 2.5	2.5 ± 2.2	0.959
TGC (mmol/L)	1.17 ± 0.56	0.94 ± 0.6	0.65 ± 0.32	<0.001
CHL (mmol/L)	4.5 ± 2.0	3.7 ± 2.5	2.7 ± 1.5	<0.001
HDL-C (mmol/L)	1.28 ± 0.86	0.97 ± 0.48	0.75 ± 0.7	0.002
LDL-C (mmol/L)	3.1 ± 1.4	2.4 ± 2.1	1.6 ± 1.3	<0.001

* Kruskal Wallis test, CP = Child–Pugh, MELD = model of end-stage liver disease, BMI = body mass index, FPG = fasting plasma glucose, IRI: insulin resistance index, TGC = triglycerides, CHL = cholesterol, HDL-C = high density lipoprotein cholesterol, LDL-C = low density lipoprotein cholesterol.

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
