# Peer review of "Leptin as a Key Player in Insulin Resistance of Liver Cirrhosis? A Cross-Sectional Study in Liver Transplant Candidates"

_jcm, 2020, doi:10.3390/jcm9020560_

Round 1
Reviewer 1 Report
Comments to authors
Košuta et al analized the association between clinical parameters and severity of the liver disease according to Child–Pugh classification and examined the effects of adipokine levels on insulin resistance in 126 non-diabetic cirrhotic patients. The results indicate that leptin levels can serve as a contributing factor to insulin resistance in these patients. This study is well conducted with appropriate methods. The data is presented clearly. However, I am afraid that the study is not qualified for the publication of this journal.
The authors should show biochemical liver function tests parameters for baseline characteristics in Table 2 I think that results in Figure 1 can be contained in Table 4. The clinical definition of insulin resistance remains elusive. The author should show the data of quantitative insulin sensitivity check index (QUICKI) as insulin sensitivity and discuss the contribution of insulin sensitivity or insulin resistance in clinical outcomes including, hepatic decompensation, HCC development, and liver related mortality. One would be interested in the correlation of fibrosis markers such as 7S-domain(7S), type III procollagen N-peptide (P3P), hyaluronic acid and tissue inhibitor of metalloproteinase-1 with adipokine levels.
Author Response
Iva Košuta, MD
Intensive Care Unit, Department of Internal Medicine, UHC Zagreb
MDPI jcm Editorial Office
Zagreb, February 4th, 2020
Re: Manuscript jcm-710693 -, entitled " Leptin as a key player in insulin resistance of liver cirrhosis?: a cross-sectional study in liver transplant candidates"
Dear Editor,
Thank you for your letter of January 29th and the possibility of resubmitting our revised manuscript entitled "Leptin as a key player in insulin resistance of liver cirrhosis?: a cross-sectional study in liver transplant candidates" for consideration for publication in the Journal of Clinical Medicine. We have carefully considered the reviewers’ comments and have revised the manuscript accordingly Below are the answers to specific reviewers’ comments.
The entire manuscript has been thoroughly edited by Grammarly Premium software and a native English speaker, and the language has been polished accordingly.
Regarding Reviewer's comments and suggestions, we offer following answers:
Reviewer 1
Comments to authors
Košuta et al analized the association between clinical parameters and severity of the liver disease according to Child–Pugh classification and examined the effects of adipokine levels on insulin resistance in 126 non-diabetic cirrhotic patients. The results indicate that leptin levels can serve as a contributing factor to insulin resistance in these patients. This study is well conducted with appropriate methods. The data is presented clearly. However, I am afraid that the study is not qualified for the publication of this journal.
The authors should show biochemical liver function tests parameters for baseline characteristics in Table 2
A: The biochemical liver function tests parameters have been added to Table 2
I think that results in Figure 1 can be contained in Table 4.
A: This is correct. We apologize for this omission which led to an over-presentation of data. The results of adiponectin and leptin has been deleted from Table 4 and presented in Figure 1.
The clinical definition of insulin resistance remains elusive.
A: The rationale for applied cut-off is described in more detail now (lines 148-153). A respective reference has been listed at No 35. The cut-off used was established as the upper 75th percentile of HOMA2-IR in a validated study with a matching Central-European population, as described in a validated study. Considering the lack of harmonization in IRI cut-off due to biological and analytical variability, an evidence-based cut-off on a continuum of insulin resistance index variable considered as optimal determinant for metabolic syndrome was applied.
The author should show the data of quantitative insulin sensitivity check index (QUICKI) as insulin sensitivity and discuss the contribution of insulin sensitivity or insulin resistance in clinical outcomes including, hepatic decompensation, HCC development, and liver related mortality.
A: The Quantitative Insulin Sensitivity Check Index (QUICKI) is a surrogate measure of insulin sensitivity derived from fasting plasma glucose (FPG) and insulin (FPI) concentration. QUICKI is identical to the simple equation form of the original HOMA1-model in all comparative respects, except that QUICKI uses a log transformed values for glucose and insulin.
QUICKI = 1/[logFPI=+logFPG]= 1/[log(FPI*FPG)]
Thus, QUICKI is simply a log-transformed value of HOMA1-IR, with the same advantages and limitations. On the other hand, an updated HOMA2 calculator, used in this study, provides reliable estimates of model parameters for fasting plasma glucose concentration up to 20.0 mmol/L and specific insulin concentration ranging from 20-300 pmol/L. HOMA2 model is a non-liner model accouning for variations in hepatic and peripheral glucose resistance, as well as circulating proinsulin levels. A value for insulin sensitivity expressed as HOMA2-%S can also be obtained from this model, but this parameter is simply the reciprocal of HOMA2-IR. Thus, no advantage of using QUICKI over HOMA2-IR could be expected in this study.
HOMA2-model is described now in detail in the new section of the Methods „Estimate of insulin resistance“, lines 130-142, while the contribution of insulin sensitivity/resistance to hepatic clinical outcomes has been addressed throughout Introduction section.
One would be interested in the correlation of fibrosis markers such as 7S-domain(7S), type III procollagen N-peptide (P3P), hyaluronic acid and tissue inhibitor of metalloproteinase-1 with adipokine levels.
A: This is certainly an interesting issue and a valid research goal. However, this was not the aim of our study. However, we have emphasized the divergent roles of adiponectin and leptin in liver fibrosis in the Discussion section.
In conclusion, we thank the reviewers for recognizing the presented study as a good scientific effort, and appreciate the comments which helped us make the article more clear and focused. We hope that we have improved the consistency, clarity and interpretation of data in the revised manuscript, with special emphasis on the discussion. We hope that the revised manuscript will meet the reviewers’ requirements and be suitable for publication in the Journal of Clinical Medicine journal.
Sincerely,
Iva Košuta, MD
Reviewer 2 Report
The authors described to investigate the association between adiponectin and leptin with insulin resistance in cirrhotic patients and to assess the influence of disease severity on insulin resistance and metabolic status. This is unique and well organized study and results are interesting. However, there are a few concerned points.
Title
The author described that there is strong correlation between Leptin and insulin resistance. However, they didn’t described Leptin as a key player in insulin resistance. The authors should explain why is Leptin a key player.
Abstract
Why didn’t authors mention regarding to triglycerides in conclusion?
Introduction
The authors confused mechanism of liver cirrhosis and metabolic status such as;
Type 2 diabetes and cardiovascular disease occur due to hyperglycemia not increasing of insulin secretion.
Fibrosis occur not due to IR due to inflammation in hepatocyte such as steatohepatitis.
Insulin receptor desensitization and down regulation occur due to protect hypoglycemia from hyperinsulinemia. etc.
Patients and Methods (Study group)
There are 14 patients with Child A. Are they candidate of liver transplantation?
Patients and Methods (Exclusion criteisa)
Why did patients with prior-Solid organ transplantation exclude? The author explained that the study patients are candidate of liver transplantation.
Results
Table 2: What did the data of total population mean?
Results
How to use the serum levels of leptin? The authors should explain the cut-off levels of serum leptin and mechanism of leptin on insulin resistance.
Discussion
The authors should explain the leptin as a key player in insulin resistance.
Conclusion
Elevated leptin; a marker of leptin resistance……. What does it mean?
Author Response
Iva Košuta, MD
Intensive Care Unit, Department of Internal Medicine, UHC Zagreb
MDPI jcm Editorial Office
Zagreb, February 4th, 2020
Re: Manuscript jcm-710693 -, entitled " Leptin as a key player in insulin resistance of liver cirrhosis?: a cross-sectional study in liver transplant candidates"
Dear Editor,
Thank you for your letter of January 29th and the possibility of resubmitting our revised manuscript entitled "Leptin as a key player in insulin resistance of liver cirrhosis?: a cross-sectional study in liver transplant candidates" for consideration for publication in the Journal of Clinical Medicine. We have carefully considered the reviewers’ comments and have revised the manuscript accordingly Below are the answers to specific reviewers’ comments.
The entire manuscript has been thoroughly edited by Grammarly Premium software and a native English speaker, and the language has been polished accordingly.
Regarding Reviewer's comments and suggestions, we offer following answers:
Reviewer 2
The authors described to investigate the association between adiponectin and leptin with insulin resistance in cirrhotic patients and to assess the influence of disease severity on insulin resistance and metabolic status. This is unique and well organized study and results are interesting. However, there are a few concerned points.
Title
The author described that there is strong correlation between Leptin and insulin resistance. However, they didn’t described Leptin as a key player in insulin resistance. The authors should explain why is Leptin a key player.
A: Our study offers some new insights into a complex and yet unclear role of various metabolic pathways associated with insulin resistance in liver cirrhosis, with possibly essential role of leptin. As mentioned as one of the limitations, cross-sectional design of our study didn't allow to conclude about causal relationships. There was a question-mark in the original title, which was lapsed during submission. We have amended it in the revised version.
Abstract
Why didn’t authors mention regarding to triglycerides in conclusion?
A: Amended in the revised version. Thank you.
Introduction
The authors confused mechanism of liver cirrhosis and metabolic status such as;
Type 2 diabetes and cardiovascular disease occur due to hyperglycemia not increasing of insulin secretion.
Fibrosis occur not due to IR due to inflammation in hepatocyte such as steatohepatitis.
Insulin receptor desensitization and down regulation occur due to protect hypoglycemia from hyperinsulinemia. etc.
A: We have re-phrased Introduction section in order to amend this misunderstanding.
Patients and Methods (Study group)
There are 14 patients with Child A. Are they candidate of liver transplantation?
A: Not yet, but they have been evaluated for LT as a plausible treatment option pro futuro.
Patients and Methods (Exclusion criteria)
Why did patients with prior-Solid organ transplantation exclude? The author explained that the study patients are candidate of liver transplantation.
A: We have investigated subjects with liver cirrhosis. Previous solid organ transplantation would involve patients with diverse hepatic and extra-hepatic comorbidities, treated with immunosuppressants. All of this would compromise the integrity of results, so previous organ transplantation was regarded as an exclusion criterion.
Results
Table 2: What did the data of total population mean?
A: All of the subjects included in the study.
Results
How to use the serum levels of leptin? The authors should explain the cut-off levels of serum leptin and mechanism of leptin on insulin resistance.
A: Actually, there is no a cut-off value for leptin in our manuscript. Regarding lack of standardization of leptin assays, it would be difficult to propose a single cut-off value depicting leptin-sensitive vs. leptin-resistant individuals. Not to mention that a putative cut-off would be population-dependent as well. Rather, our study sought to investigate the association between adiponectin and leptin with insulin resistance in patients with liver cirrhosis, as well as to assess the influence of disease severity on insulin resistance and metabolic status. Despite the limitations, our study offers some new insights into a complex and yet unclear role of various metabolic pathways associated with insulin resistance in liver cirrhosis. Further research is needed to determine causal relationships between the observed metabolic factors and clinical outcomes of patients with liver cirrhosis. Please refer to Introduction and discussion sections for the explanation.
Discussion
The authors should explain the leptin as a key player in insulin resistance.
A: Re-phrased and explained throughout Discussion section
Conclusion
Elevated leptin; a marker of leptin resistance……. What does it mean?
A: As above
In conclusion, we thank the reviewers for recognizing the presented study as a good scientific effort, and appreciate the comments which helped us make the article more clear and focused. We hope that we have improved the consistency, clarity and interpretation of data in the revised manuscript, with special emphasis on the discussion. We hope that the revised manuscript will meet the reviewers’ requirements and be suitable for publication in the Journal of Clinical Medicine journal.
Sincerely,
Iva Košuta, MD
Round 2
Reviewer 1 Report
I still I think that results in Figure 1 should be deleted and contained in Table 4. I have no further comment to make.
Author Response
The graph containing the distribution of leptin through the Child-Pough stages has been omitted, and the results are now contained in table 4.
Reviewer 2 Report
The authors corrected for reviewer's requests point-by-point. And the manuscript became clear to conclude.
Author Response
No additional changes have been made.